# Middle Cerebral Artery Doppler Velocimetry for the Diagnosis of Twin Anemia Polycythemia Sequence: A Systematic Review

**DOI:** 10.3390/jcm9061735

**Published:** 2020-06-04

**Authors:** Clifton O. Brock, Eric P. Bergh, Kenneth J. Moise,, Anthony Johnson, Edgar Hernandez-Andrade, Dejian Lai, Ramesha Papanna

**Affiliations:** 1Department of Obstetrics, Gynecology, and Reproductive Sciences, McGovern Medical School at the University of Texas Health Science Center at Houston, 6431 Fannin Street, Houston, TX 77030, USA; clifton.o.brock@uth.tmc.edu (C.O.B.); Eric.P.Bergh@uth.tmc.edu (E.P.B.); kenneth.j.moise@uth.tmc.edu (K.J.M.J.); anthony.johnson@uth.tmc.edu (A.J.); edgar.a.hernandezandrade@uth.tmc.edu (E.H.-A.); 2The Fetal Center Children’s Memorial Hermann Hospital, Houston, TX 77030, USA; 3Division of Biostatistics, University of Texas, School of Public Health, 1200 Pressler St, Houston, TX 77030, USA; Dejian.lai@uth.tmc.edu

**Keywords:** twin anemia polycythemia sequence, TAPS, monochorionic diamniotic twins, prenatal ultrasound

## Abstract

Twin anemia polycythemia sequence (TAPS) is a rare complication of monochorionic diamniotic (MCDA) twins. Middle cerebral artery peak systolic velocity (MCA-PSV) measurements are used to screen for TAPS while fetal or neonatal hemoglobin levels are required for definitive diagnosis. We sought to perform a systematic review of the efficacy of MCA-PSV in diagnosing TAPS. Search criteria were developed using relevant terms to query the Pubmed, Embase, and SCOPUS electronic databases. Publications reporting diagnostic characteristics of MCA-PSV measurements (i.e., sensitivity, specificity or receiver operator curves) were included. Each article was assessed for bias using the Quality Assessment of Diagnostic Accuracy Studies II (QUADAS II) tool. Results were assessed for uniformity to determine whether meta-analysis was feasible. Data were presented in tabular form. Among publications, five met the inclusion criteria. QUADAS II analysis revealed that four of the publications were highly likely to have bias in multiple areas. Meta-analysis was precluded by non-uniformity between definitions of TAPS by MCA-PSV and neonatal or fetal hemoglobin levels. High-quality prospective studies with consistent definitions and ultrasound surveillance protocols are still required to determine the efficacy of MCA-PSV in diagnosing TAPS. Other ultrasound findings (e.g., placenta echogenicity discordance) may augment Doppler studies.

## 1. Introduction

Twin Anemia-Polycythemia Sequence (TAPS) is a rare complication of monochorionic diamniotic (MCDA) twin pregnancies first reported following fetoscopic laser surgery (FLS) for twin to twin transfusion syndrome (TTTS) in 2006 [1]. TAPS is caused by chronic blood transfusion through shared placental vessels leading to substantial morbidity as the twins become increasingly anemic and plethoric. While the disease may follow FLS (referred to here as “post-laser TAPS” (plTAPS)) in 3–10% of MCDA pregnancies, spontaneous TAPS (sTAPS), first described in 2007 by Lopriore et al., may develop 2–5% of the time [2,3,4,5,6,7,8]. Middle cerebral artery peak systolic velocity (MCA-PSV), as measured during Doppler velocimetry studies, is used to screen for TAPS. Absolute criteria (i.e., MCA-PSV < 1.0 MoMs in the plethoric twin plus MCA-PSV > 1.5 MoMs in the anemic twin, MoM = multiples of the median) and relative criteria (e.g., ΔMCA-PSV > 0.373 or > 0.5 MoMs) have been proposed for diagnosis [5,9,10]. Definitive diagnosis is obtained by comparing twin serum hemoglobin (Hb) levels with or without reticulocyte count ratios either postnatally or by intrauterine umbilical cord blood sampling. Treatment modalities vary by gestational age (GA) and include expectant management, fetoscopic laser surgery, intrauterine transfusion (IUT) with or without partial exchange transfusion (PET), selective reduction and preterm delivery [11,12].

Routine use of MCA Doppler velocimetry to detect TAPS is currently recommended by the International Society for Ultrasound in Obstetrics and Gynecology (ISUOG) [13]. However, in 2013 the Society for Maternal Fetal Medicine (SMFM) recommended against this practice stating that there is “no evidence that monitoring for TAPS with MCA PSV Doppler at any time, including >26 weeks, improves outcomes [14]”. Since 2013, TAPS has received considerable attention in the literature and the first randomized clinical trial for treatment of TAPS is currently underway [15]. Following these developments, there has been increased enthusiasm for screening protocols utilizing serial MCA Doppler velocimetry [16].

MCA Doppler velocimetry is also the principle screening test for fetal anemia in the setting of maternal alloimmunization to fetal red blood cell antigens [17]. The use of MCA-PSV values to screen for fetal anemia from alloimmunization has been established following extensive research, but with a false positive rate of 10% still prevailing [18]. The method, which may be affected by a range of physiologic parameters, is less well studied in TAPS, which is a unique physiologic entity. Furthermore, MCA Doppler velocimetry may be less efficacious for detecting polycythemia than for anemia [19]. Finally, the definitive diagnosis of TAPS and verification of MCA Doppler velocimetry as a screening tool is confounded at early GAs by the inability to sample fetal Hb due to technical limitations and concern regarding perinatal risks. In this study, we sought to exhaustively review the literature to determine the diagnostic efficacy of MCA Doppler velocimetry in screening for TAPS.

## 2. Materials and Methods

We performed a systematic review following PRISMA guidelines [20]. The research question of interest was to determine the sensitivity, specificity, positive predictive value, and negative predictive value of MCA Doppler velocimetry for antenatal diagnosis of TAPS. Eligible studies included those where MCA-PSV values were collected to screen for TAPS and fetal or neonatal serum Hb concentrations were also collected as the “gold standard” diagnostic test. Studies written in English during or after 2006 (i.e., when TAPS was first described in the literature) were considered [1]. Case reports, case series and literature that did not undergo peer review (i.e., abstracts, clinical commentary) were excluded. 

To locate eligible studies, a search strategy was developed and applied to the Pubmed, Embase and SCOPUS electronic databases. The following terms were queried in April of 2020: “twin anemia polycythemia sequence,” “TAPS” and “feto-fetal transfusion.” In the SCOPUS search, only publications in the “Medicine” and “Health Professions” subject areas were included. Abstracts for all publications retrieved by this search were reviewed by two of the authors (C.B. and E.B.). Publications that appeared as though they may include antenatal screening for TAPS based on abstract review were obtained for complete review. A secondary search (i.e., “by hand”) was performed by reviewing the bibliographies of these publications. The aim of this process was to generate a comprehensive list of primary studies suitable to answer the research question.

Documents retrieved by the primary and secondary searches were reviewed in their entirety. Two independent reviewers (C.B. and E.B.) carefully examined each document and extracted relevant data including the author, PMID number, year of publication, journal and study type (i.e., prospective vs. retrospective). Studies that compared antenatal screening for TAPS by MCA-PSV values to fetal or neonatal serum Hb levels, including either sensitivity/specificity calculations or derivation of a receiver-operative curve (ROC) were considered for analysis. For these studies, we extracted study size, number of TAPS cases, number of plTAPS and sTAPS, stage and GA at TAPS diagnosis and the date range over which women were clinically evaluated. Outcome measures including the sensitivity, specificity, positive predictive value (PPV), negative predictive value (NPV) and area under the ROC were also collected. Uniformity of the studies was assessed for the possibility of performing meta-analysis. Because determination of fetal Hb levels may not be possible for pregnancies at earlier GAs, it was also noted whether calculations were stratified by GA. Any discrepancies in extracted data or study inclusion were arbitrated and resolved by a third author (R.P.).

The Quality Assessment of Diagnostic Accuracy Studies II (QUADAS II) methodology was applied to the selected studies to determine their likelihood of bias and applicability in answering the research question [21]. QUADAS II includes a structured list of 11 “signaling questions” evaluating for bias in the following four domains: (1) patient selection, (2) collection and interpretation of an index test (i.e., MCA-PSV to screen for TAPS), (3) collection and interpretation of a reference standard (i.e., Hb levels to definitively diagnose TAPS), (4) flow/timing of the study. Questions may be answered “yes,” “no,” or “unclear.” Additional questions may be added at the users’ discretion to address possible biases unique to the research question. We added two questions: (1) “Was a clearly defined schedule described for timing of initial and follow-up application of the index test?” (2) “Were reference test values from repeat intrauterine transfusions or reference test values following birth in neonates who had a previous intrauterine transfusion used to calculate sensitivity/specificity?” MCA-PSV is usually measured serially as recommended by ISUOG; the first question was added because a missed diagnosis following inadequate antenatal screening may lead to underestimation of the test’s sensitivity. The second question was added for two reasons: (1) Hb values at the time of intervention, not postnatal Hb values, should be compared with MCA-PSV values at the time of TAPS diagnosis for fetuses that underwent transfusion. (2) The efficacy of MCA-PSV values in predicting fetal anemia decreases following intrauterine transfusion [22]. Based on answers to these 13 questions (see Table 1), the likelihood of bias in the four domains was assessed as “HIGH,” “LOW,” or “UNCLEAR.” Similarly, but without use of signaling questions, concerns about applicability of the studies to the four domains are assessed as “HIGH,” “LOW,” or “UNCLEAR.”

The aim of this study was to collate and summarize the results of primary studies and, if possible, perform meta-analysis to estimate the pooled sensitivity and specificity of MCA-PSV values for diagnosing TAPS. Data were summarized in tabular form. Microsoft Excel (Redmond, WA) was used to tabulate search results, extracted data, and QUADAS II results.

## 3. Results

The primary search of the Pubmed, Embase, and SCOPUS databases produced 432 publications. Based on abstract review, 54 of these publications were selected to be reviewed in their entirety [1,2,4,5,7,9,10,11,12,16,19,23,24,25,26,27,28,29,30,31,32,33,34,35,36,37,38,39,40,41,42,43,44,45,46,47,48,49,50,51,52,53,54,55,56,57,58,59,60,61,62,63,64,65]. The secondary search of bibliographies produced three additional publications which were also reviewed in their entirety for a total 57 publications [6,66,67]. The publications are detailed in Table 1 and reasons for exclusion are summarized in Figure 1. Five of the 57 publications, all from the primary search, described a study of MCDA twins where serial ultrasounds with MCA-PSV values were collected to detect TAPS with direct comparison to Hb values [5,9,10,19,47]. 

After applying QUADAS II methodology to these five studies, four were determined highly likely to have bias in patient selection (Figure 2) [5,9,10,47]. Three were found highly likely to have bias in conduction or interpretation of the index test (i.e., MCA-PSV Doppler values) as well as conduction or interpretation of the standard reference (i.e., Hb levels) [5,10,47]. All of the studies were determined to have low likelihood of bias in flow and timing. The answers to the QUADAS II signaling questions provide rationale for these assessments of bias and are detailed in Table 2. Regarding patient selection, studies by Slaghekke et al. and Veujoz et al. only included women with a diagnosis of TAPS without any uncomplicated MCDA twins [9,47]. Hence, the number of true negatives and false positives, or women without TAPS, appear to have been based on women who were ultimately diagnosed with TAPS with MCA-PSV values and Hb levels collected prior to diagnosis or after treatment and resolution of the disease. Studies by De Sousa et al. and Tollenaar et al. excluded twins that were missing either MCA-PCV values or Hb levels [5,10]. As protocols for serial MCA-PSV analysis (or compliance with such a protocol) over the study periods were not available, the reasons these values were missing, and women excluded, could not be determined. One study appeared to include postnatal Hb levels to calculate sensitivity, specificity, PPV, and NPV in twins that had a prior IUT and another used MCA-PSV/Hb pairs from the time of repeat IUT for the same purpose [10,47]. None of the studies were found to have concerns regarding the applicability of patient selection, the index test, or the standard reference. 

Figure 2 Legend: Results of QUADAS II application. QUADAS II consists of four assessments of bias and three assessments of applicability to the study question. A judgement of “HIGH,” “LOW,” or “UNCLEAR” is made for each assessment of each study considered. A “HIGH” judgement means that the study methodology is likely to introduce bias (for bias assessments) or likely lacks applicability to the study question (for applicability assessments). A “LOW” means there is less likelihood of bias or lack of applicability. Answers to the “signaling questions” which support the bias judgements which are detailed in Table 3.

Demographic characteristics of women from each of the selected studies are detailed in Table 3. The studies included 779 women and 122 cases of TAPS. There may be overlap in TAPS cases between studies by Tollenaar et al. and Slaghekke et al. as they are from the same institution (Leiden University, Netherlands) with overlapping study periods [10,47]. All but one study (Slaghekke et al.) reported that the incidence of TAPS ranged from 4.6% to 35.3% (sTAPS and plTAPS together). Tollenaar et al. did not report the incidence of plTAPS and sTAPS separately for all cases (see fotenote, Table 3) [10]. Among the remaining studies, the incidences of plTAPS (total *n* = 44) and sTAPS (total *n* = 43) were reported separately, but each type was included together in calculations of sensitivity, specificity, PPV and NPV. Among studies where plTAPS and sTAPS were delineated and incidence was reported, the combined incidence of sTAPS was 4.8% (31/643). The majority of TAPS was diagnosed after 24 weeks with delivery (mostly by Cesarean section) in the late 3^rd^ trimester (i.e., >32 weeks). Only Veujoz et al. described the disease severity by stages [9]. 

Table 4 shows sensitivity, specificity, PPV, and NPV of MCA-PSV values in predicting the diagnosis of TAPS by differences in intertwin Hb levels. All studies limited analysis to patients where an MCA-PSV value was collected within one week of determining Hb levels by cordocentesis (antenatally) or venipuncture (postnatally). Studies by Tollenaar et al. and Fishel-Bartal et al. defined prenatal TAPS with absolute values (i.e., MCA-PSV > 1.5 MoMs and MCA-PSV < 1.0 MoMs) as well as the relative difference between values [10,19]. De Sousa et al. considered only relative values, and the remaining two studies used only absolute values [5,9,47]. Collectively, four different criteria for a positive MCA-PSV screening test were used between the five studies. None of the studies stratified calculations of the diagnostic characteristics of MCA-PSV values by GA or treatment received. 

The criteria for definitive diagnosis of TAPS was similarly heterogeneous. Each study included numerical cutoffs for diagnosis; however, three evaluated the difference in Hb levels between twins for diagnosis of TAPS, while the remaining two used individual Hb levels to diagnose anemia and polycythemia separately. Two of the studies also required the reticulocyte count ratio of the polycythemic twin to the anemic twin to be greater than 1.7 [9,10]. Collectively, the five studies had four different definitions of definitive TAPS. The sensitivities, specificities, PPVs, and NPVs were reported for four of the five studies, while Bartal et al. reported ROC characteristics [19]. Finally, none of the studies stratified calculations by GA or intrauterine treatment for any of the calculations.

Heterogeneity between the studies precluded meta-analysis for pooled estimates of PPV, NPV, sensitivity, or specificity.

## 4. Discussion

This systematic review arrived at several important findings on the efficacy of MCA-PSV values in diagnosing TAPS: (1) Five studies (two prospective, three retrospective) were found that describe the diagnostic characteristics of MCA-PSV values. (2) plTAPS and sTAPS were considered together in calculations of diagnostic characteristics for all studies. (3) No study stratified calculations of the diagnostic characteristics by GA or treatment. (4) Diagnostic criteria (i.e., “cutoff values”) for the screening test and the diagnostic test were not consistently defined between studies. (5) Because of their retrospective nature, most of the studies likely exhibit bias in patient selection and conduction/interpretation of the index test (MCA-PSV values) and standard reference (Hb levels).

The above findings suggest significant limitations in the diagnosis and management of TAPS. First, false positive and false negative diagnoses are of major concern. Currently, fetal intervention may be undertaken based on MCA Doppler velocimetry which appears to have PPVs ranging from 70% to 100%; however, these PPVs are estimated in retrospective studies with a high likelihood of bias, particularly in patient selection [5,10,47]. This is not surprising as each study includes data from before 2016, when ISUOG first recommended universal serial MCA Doppler velocimetry in MCDA twins [13]. Prospective data by Fishel-Bartal et al. suggest that absolute MCA-PSV values perform modestly in diagnosing TAPS (AUC = 0.687, 95% CI (0.547–0.827) for anemia and AUC = 0.617, 95% CI (0.505–0.728) for polycythemia) [19]. While the same study shows better performance for relative MCA-PSV values (i.e., ΔMCA-PSV, AUC = 0.871, 95% CI (0.757–0.985)), these data are yet to be corroborated in another high-quality prospective study. Furthermore, the incidence of sTAPS in this study (10.1%) is more than twice as high as reported elsewhere (1.2–4.9%, except for De Sousa et al. with an incidence of 9.1%) increasing the likelihood that PPV is overestimated [2,3,4,5,6,7]. Hence, the risk of intervention for TAPS when disease is not truly present may be substantial. As the stage of TAPS is only reported in one of the studies, the degree to which MCA-PSV values may over- or underestimate severe outcomes, which provide greater impetus for intervention, is also unclear. 

Of note, the work by Fishel-Bartal et al. highlights an emerging trend of favoring ΔMCA-PSV over absolute MCA-PSV values because of the former’s reported higher sensitivity [19]. Tollenaar et al. retrospectively observed that twins meeting ΔMCA-PSV criteria, but not absolute criteria, have similar postnatal outcomes to twins meeting absolute criteria. This led that group to propose a new staging system for TAPS based on ΔMCA-PSV [10]. Fishel-Bartal et al. suggest that the superiority of ΔMCA-PSV values may be related to the poor predictive ability of MCA-PSV < 1.0 MoM in diagnosing polycythemia; however, Slaghekke et al. report high sensitivity and specificity using this method (Table 4) [19,47]. Further prospective data are required to validate (or invalidate) MCA-PSV < 1.0 MoM and ΔMCA-PSV in diagnosing polycythemia and TAPS, respectively.

The evaluation of diagnostic performance is further complicated by the inability to measure Hb levels when TAPS is diagnosed at an early GA or following FLS. The treatment algorithm published in Leiden recommends FLS prior to 28 weeks, preterm delivery after 32 weeks, and IUT with PET at intermediate GAs for Stage II or greater TAPS [42]. At GAs greater than 32 weeks, MCV-PSV values and Hb levels may be measured in close temporal proximity because delivery is likely to occur soon regardless of whether TAPS is diagnosed (Figure 3A). Thus, the incidence of all four possible outcomes relating MCA-PSV to Hb (true and false positives, true and false negatives) may be determined. These values are required to determine sensitivity, specificity, PPV, and NPV. 

For GAs of 28 to 32 weeks, Hb would only be measured when IUT/PET is performed in the setting of Stage II or greater TAPS. As such, true and false negatives cannot be determined (Figure 3B). It is possible that undiagnosed TAPS (i.e., false negatives) account for some portion of the fetal demise observed in otherwise normal appearing MCDA twins [68,69,70]. For GA less than 28 weeks, none of the four possible outcomes may be determined because Hb levels are not typically measured in the setting of FLS (Figure 3C). The diagnostic efficacy of MCA-PSV values cannot be determined if the gold standard diagnostic test (i.e., Hb levels) is unavailable. Using one or more surrogate markers (a so called “gold alloy”) for diagnosis, such as the “starry liver sign”, discordant placental echogenicity, high-output cardiac failure or polycythemic thrombosis (i.e., fetal limb necrosis, bowel infarction with perforation or intracranial hemorrhage) is one method to address such diagnostic quandaries; however, use of this approach was not encountered in our search of the literature [29,71,72].

Following diagnosis, our understanding of the role for various intrauterine treatments is evolving. A recent multicenter study (*n* = 370) reports a high level of variance in the treatments used (i.e., FLS, IUT/PET, expectant management, early delivery and selective feticide) and the GAs at which they are employed. An advantage in pregnancy prolongation is reported for FLS, but comparative treatments were performed at later GAs [73]. Several smaller studies also address treatment (total *n* = 185); however, each combined plTAPS with sTAPS, had unsystematic methods of patient inclusion, and directly compared treatments performed across the full range of gestational ages [11,26,52]. 

Further work is required to provide a valid basis for the diagnosis and treatment of TAPS. sTAPS and plTAPS should be considered separately as data is sparse to suggest whether MCA-PSV values behave similarly in these two physiologically distinct scenarios. We are unaware of any data that separate the two entities as part of an evaluation of MCA-PSV as a screening test. Reliable false positive rates for MCA-PSV may help determine whether the risk of intervention is appropriate. At early GAs, surrogate markers for the disease will be required to determine estimates for the incidence of false positive diagnosis. Further work is also required to elucidate the benefits of intrauterine treatment. While grave outcomes such as neonatal death, ischemic loss of fetal limbs, and cerebellar disruption have been described, it is unclear whether a treatment protocol based on MCA-PSV values can prevent these outcomes [1,74,75,76].

In summary, this systematic review of MCA-PSV values for diagnosis of TAPS reveals that studies supporting this methodology are dissimilar and at risk for bias. The resulting estimates of the PPV of MCA-PSV measurements may be unreliable, and thus lead to attempted treatment of normal twins, particularly in the setting of Stage II disease. Further work is required to reliably estimate the diagnostic performance of MCA-PSV values and determine the risks and benefits of intrauterine treatment of TAPS. Therefore, in the current practice with limited strong supporting data, it is pragmatic to continue surveillance with antenatal monitoring of MCA-PSV for MCDA twins, but to interpret the results with caution. Intervention should be more strongly considered when additional abnormalities are noted such as cardiac failure, abnormal Doppler flow in umbilical artery or ductus venosus, discordant placental echogenicity or hydrops. Even in the presence of normal MCA-PSV in MCDA twins, it is prudent to consider evaluating for other signs of TAPS during ultrasound surveillance. Until more rigorous, large, prospective and well-controlled studies are conducted, many questions related to screening for TAPS will remain unanswered.

## Figures and Tables

**Figure 1 jcm-09-01735-f001:**
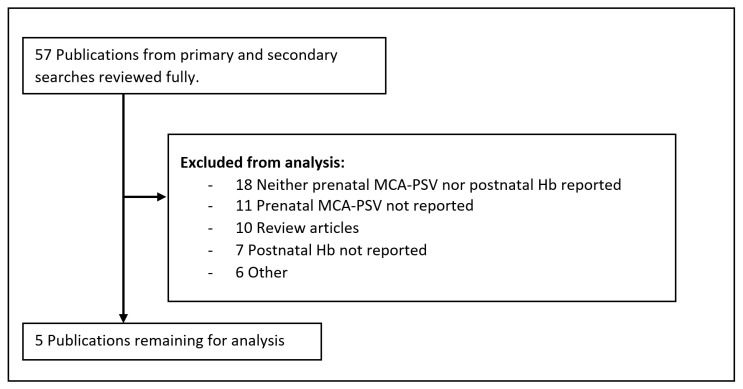
Flowchart of Study Inclusion.

**Figure 2 jcm-09-01735-f002:**
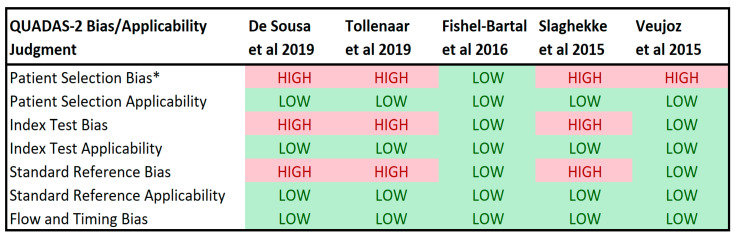
QUADAS II Systematic Assessment of Bias for Included Studies.

**Figure 3 jcm-09-01735-f003:**
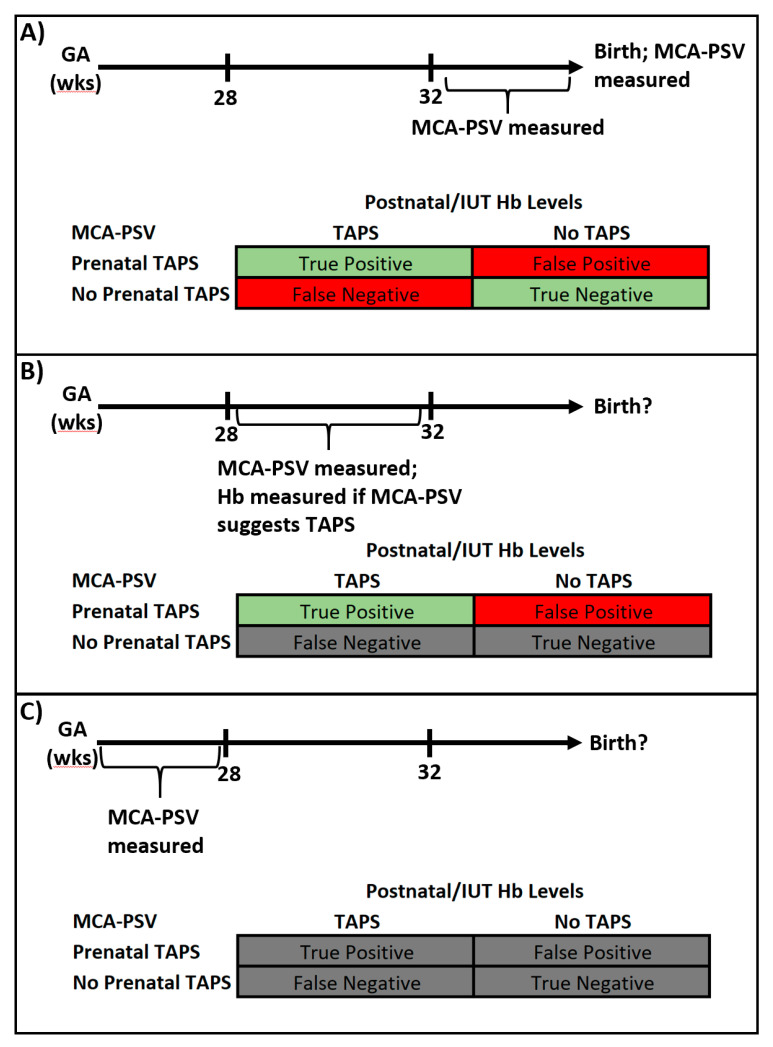
TAPS Screening by Gestational Age (or Treatment Modality). Ability to determine true or false positives and negatives according to the availability of Hb levels (i.e., the gold standard) at (**A**) greater than 32 weeks, (**B**) 28–32 weeks, and (**C**) less than 28 weeks gestational age. Values shaded in red or green can be determined whereas values shaded in grey cannot be determined.

**Table 1 jcm-09-01735-t001:** Studies Select for Full Review for MCA-PSV diagnosis of TAPS.

Document Number	Author	Pubmed ID	Year	Reason for Exclusion
1	Visser et al.	31261154	2020	Prenatal MCA-PSV not reported
2	Tollenaar et al.	31432580	2020	Postnatal Hb not reported
3	Mao et al.	32056557	2020	Neither prenatal MCA-PSV nor postnatal Hb were reported
4	Tavares de Sousa et al.	30207009	2019	Included
5	Tollenaar et al.	30125414	2019	Included
6	Nicholas et al.	30763267	2019	Review article
7	Hill et al.	31230662	2019	Systematic review on treatments
8	Nicholas et al.	31277521	2019	Review article
9	Tollenaar et al.	31366031	2019	Postnatal Hb not reported
10	Khalil et al.	31605505	2019	Review article
11	Tollenaar et al.	31856326	2019	Postnatal Hb not reported
12	Bamberg et al.	28557152	2018	Postnatal Hb not reported
13	Donepudi et al.	28925589	2018	Postnatal Hb not reported
14	McDonald et al.	27561094	2017	Neither prenatal MCA-PSV nor postnatal Hb were reported
15	Verbeek et al.	28460542	2017	Prenatal MCA-PSV not reported
16	De Paepe et al.	29208240	2017	Prenatal MCA-PSV not reported
17	Fishel-Bartal et al.	26663574	2016	Included
18	Sato et al.	26760043	2016	Study on triplets with monochorionic placentation
19	Suzuki et al.	27257402	2016	Neither prenatal MCA-PSV nor postnatal Hb were reported
20	Lucewicz et al.	25580896	2016	Review article
21	Donepudi et al.	26033705	2016	Postnatal Hb not reported
22	Stagnati et al.	26173065	2016	Postnatal Hb not reported
23	Ashwal et al.	26580546	2016	Neither prenatal MCA-PSV nor postnatal Hb were reported
24	Tollenaar et al.	26788848	2016	Prenatal MCA-PSV not reported
25	Tollenaar et al.	27068715	2016	Review article
26	Couck et al.	27098457	2016	Review article
27	Hiersch et al.	27734505	2016	Both parameters reported, however cannot determine true/false negatives (i.e., Hb levels when MCA-PSV were negative). Specificity, sensitivity, ROC not reported.
28	Sananes et al.	25790745	2015	Prenatal MCA-PSV not reported
29	Veujoz et al.	25484182	2015	Included
30	Yokouchi et al.	25510181	2015	Prenatal MCA-PSV not reported
31	Van Winden et al.	25721425	2015	Neither prenatal MCA-PSV nor postnatal Hb were reported
32	Slaghekke et al.	26094734	2015	Included
33	Sueters et al.	24433823	2014	Review article
34	Mabuchi et al.	24585685	2014	Prenatal MCA-PSV not reported
35	Slaghekke et al.	24706478	2014	Prenatal MCA-PSV not reported
36	Slaghekke et al.	24753027	2014	Neither prenatal MCA-PSV nor postnatal Hb were reported
37	Baschat et al.	24858317	2014	Review article
38	Rossi et al.	25222165	2014	Systematic review of outcomes
39	Zhao et al.	25449031	2014	Neither prenatal MCA-PSV nor postnatal Hb were reported
40	de Villiers et al.	23481221	2013	Neither prenatal MCA-PSV nor postnatal Hb were reported
41	Verbeek et al.	23485951	2013	Prenatal MCA-PSV not reported
42	Zhao et al.	23639577	2013	Neither prenatal MCA-PSV nor postnatal Hb were reported
43	Favre et al.	23744723	2013	Neither prenatal MCA-PSV nor postnatal Hb were reported
44	Nakayama et al.	22413750	2012	Neither prenatal MCA-PSV nor postnatal Hb were reported
45	de Villiers et al.	22257747	2012	Neither prenatal MCA-PSV nor postnatal Hb were reported
46	Lopriore et al.	21912373	2011	Neither prenatal MCA-PSV nor postnatal Hb were reported
47	Lopriore et al.	20087909	2010	Prenatal MCA-PSV not reported
48	Slaghekke et al.	20339296	2010	Review article
49	Lopriore et al.	20417489	2010	Prenatal MCA-PSV not reported
50	Habli et al.	19788973	2009	Neither prenatal MCA-PSV nor postnatal Hb were reported
51	Slaghekke et al.	20099211	2009	Review article
52	Lopriore et al.	18827116	2008	Neither prenatal MCA-PSV nor postnatal Hb were reported
53	Lopriore et al.	17877693	2007	Letter to editor
54	Robyr et al.	16522415	2006	Unclear if Hb was measured on cases with normal MCA
**Studies from the secondary search of bibliographies**				
1	Lopriore et al.	16644009	2006	Neither prenatal MCA-PSV nor postnatal Hb were reported
2	Lewi et al.	18533114	2008	Neither prenatal MCA-PSV nor postnatal Hb were reported
3	Mackie et al.	28549467	2017	Neither prenatal MCA-PSV nor postnatal Hb were reported

Abbreviations: Hb = hemoglobin concentration, MCA-PSV = middle cerebral artery peak systolic velocity.

**Table 2 jcm-09-01735-t002:** QUADAS II Signaling Questions for Included Studies.

QUADAS II Signaling Questions	De Sousa 2019	Tollenaar 2019	Fishel-Bartal 2016	Slaghekke 2015	Veujoz 2015
Domain 1: Patient Selection Risk of Bias					
Was a consecutive or random sample of patients enrolled?	No *	Yes	Yes	No ^ƚ^	No ^ƚ^
Was a case-control design avoided?	Yes	Yes	Yes	No ^ƚ^	No ^ƚ^
Did the study avoid inappropriate exclusions?	Yes	No ^ǂ^	Yes	Yes	Yes
Domain 2: Index Test Bias Risk of Bias					
Were the index test results interpreted without knowledge of the results of the reference standard?	Yes	Yes	Yes	Yes	Yes
If a threshold was used, was it prespecified?	Unclear	Unclear	Unclear	Yes	Yes
Was a clearly defined schedule described for timing of initial and follow-up application of the index test?	No ^§^	No ^§^	Yes	No ^§^	Yes
Domain 3: Reference Test Risk of Bias					
Is the reference standard likely to correctly classify the target condition?	Yes	Yes	Yes	Yes	Yes
Were the reference standard results interpreted without knowledge of the results or the index test?	Unclear	Unclear	Unclear	Unclear	Unclear
Were reference test values from repeat intrauturine transfusions or reference test values following birth in neonates who had a previous intrauterine transfusion used to calculate sensitivity/specificity?	No	Yes ^¶^	No	Yes ^#^	No
Domain 4: Flow and Timing Risk of Bias					
Was there an appropriate interval between the test(s) and reference standard?	Yes (at birth)	Yes (≤1 wk)	Yes (≤1 wk)	Yes (<48 h)	Yes (at birth)
Did all patients receive the reference standard?	Yes	Yes	Yes	Yes	Unclear
Did patients receive the same reference standard?	Yes	Yes	Yes	Yes	Yes
Were all patients included in the analysis?	Yes	Yes	No (69/162, 43%)	Yes	No (9/20, 45%)

* A retrospective study design was applied without explanation of a standardized protocol for the timing of either the index test (MCA-PSV) or reference standard (Hb) during the retrospective period. Patients were not included when MCA-PSV (73/256, 29%) or Hb data (3/256, 1%) were not available. Patients were also not included if the interval between MCA-PSV and Hb data was greater than one week. ^ƚ^ These studies were limited to patients with TAPS (i.e., not consecutive or random MCDA twins). Sensitivity and specificity were calculated using Hb and MCA-PSV values at times when patients met, versus did not meet, the criteria for diagnosis (i.e., multiple MCA-PSV, Hb pairs per patient). Slaghekke et al. did not include patients with missing data for Hb or MCA-PSV. ^ǂ^ A retrospective study design was applied without explanation of a standardized protocol for the timing of either the index test (MCA-PSV) or reference standard (Hb) during the retrospective period. Patients were excluded when MCA-PSV (221/351, 63%) or Hb (50/351, 14%) data were not available. Patients were also excluded if the interval between MCA-PSV and Hb data was greater than one week. ^§^ These studies did not describe a schedule for application of MCA-PSV measurement (i.e., patients considered, starting gestational age and frequency of follow up) or report ultrasound findings that might have prompted more frequent Doppler studies. ^¶^ Sensitivity and specificity calculations were performed using “postnatal intertwin Hb difference,” however 10 cases of intrauterine transfusion were reported. ^#^ Sensitivity and specificity calculations were performed using MCA-PSV/Hb pairs at the time of repeat IUT.

**Table 3 jcm-09-01735-t003:** Demographic characteristics of women from the selected studies.

Characteristic	De Sousa 2019	Tollenaar 2019	Fishel-Bartal 2016	Slaghekke 2015	Veujoz 2015
Study Design	Retrospective	Retrospective	Prospective	Retrospective	Prospective
Number of women	154	80	69	43	433
Number and Type of TAPS	15 (14 s + 1 pl)	35 *	9 (7 s + 2 pl)	43 (12 s + 31 pl)	20 (10 s + 10 pl)
Incidence of TAPS	9.7%	35.3%	13.0%	N/A	4.6%
GA at delivery all twins (wks)	35 (26–39) wks	35 (33–36) controls only	33.6 (24.6–38.3)	--	--
GA at diagnosis (wks)	34.8 (26–39)	<1 wk before delivery	<1 wk before delivery	25 (19–30) Antenatal diagnosis 26.5 (19–32) Postnatal diagnosis	24.8 +/− 5.9
GA at delivery for TAPS cases (wks)	29 + 4 to 37 + 0	32 (29–34)	--	25 (19–30) Antenatal diagnosis 26.5 (19–32) Postnatal diagnosis	32.1 +/− 1.9
TAPS Stage at diagnosis	Not reported	Not reported	Not reported	Not reported	Stage I-47.0% Stage II-29.4% Stage III-17.7% Stage IV-5.9% Stage V-0.0%
Treatment of TAPS	Not reported	IUT-6 IUT + PET-4	IUT-2	Not reported	IUT-2 FLS-7
Overall Survival following TAPS diagnosis	Not reported	55/58 (94.8%)	Not reported	69/82 (84.1%)	15/20 (75.0%)
Cesarean Delivery	72.1%	Not reported	50.7%	78.9%	Not reported

Abbreviations: FLS = fetoscopic laser surgery, pl = post-laser, Hb = hemoglobin concentration, IUT = intrauterine transfusion MCA-PSV = middle cerebral artery peak systolic velocity, ΔMCA-PSV = difference in MCA-PSV between twins, MoM = multiples of the median, PET = partial exchange transfusion, s = spontaneous, TAPS = twin anemia polycythemia sequence, wk = week. * Of 35 TAPS cases, type was specified for 29: There were 7 sTAPS and 22 plTAPS.

**Table 4 jcm-09-01735-t004:** Diagnostic characteristics of TAPS by MCA-PSV compared with Hb difference.

Characteristic	De Sousa 2019	Tollenaar 2019	Fishel-Bartal 2016	Slaghekke 2015	Veujoz 2015
**Interval between MCA-PSV and Hb measurement**	2 (0–7) days	<1 week	<1 week	--	<48 h
**Antenatal finding for diagnosis of TAPS**	ΔMCA-PSV > 0.373 MOM (cutoff determined by ROC analysis)	(1) MCA-PSV < 1.0 MOM (recipient), MCA-PSV > 1.5 MOM(donor) (2) ΔMCA-PSV > 0.5 MOM	(1) MCA-PSV < 1.0 MOM (recipient), MCA-PSV > 1.5 MOM (donor) (2) ΔMCA-PSV by ROC analysis. (ROC cutoff not reported)	MCA-PSV < 1.0 MOMs (recipient), MCA-PSV > 1.5 MOMs (donor)	MCA-PSV < 1.0 MOMs (recipient), MCA-PSV > 1.5 MOMs (donor)
**Postnatal finding for diagnosis of TAPS**	Hb difference > 90% (7.25 Hb)	Hb difference > 8 g/dL and Reticulocyte Ratio > 1.7 or anastomoses < 1mm	Anemia: Hct < 45% Polycythemia: Hct > 65%	Anemia: Hct > 5 SD below mean Polycythemia: Hct > 5 SD above mean	Hb difference > 8 g/dL and Reticulocyte Ratio > 1.7 or anastomoses < 1mm
**AUC (95% CI)**	0.976 (0.935–0.993)	Not reported	1) Anemia: 0.687 (0.547–0.827) Polycythemia: 0.617 (0.505–0.728) 2) TAPS 0.871 (0.757–0.985)	Not Reported	Not Reported
**Sensitivity (95% CI)**	93.3% (68.1–99.8)	1) 46% (30–62) 2) 83% (67–92)	Not reported	Anemia: 94% (85–98) Polycythemia: 97% (87–99)	71%
**Specificity (95% CI)**	95.7% (90.8–98.4)	1) 100% (29–100) 2) 100% (92–100)	Not reported	Anemia: 74% (62–83) Polycythemia: 96% (89–99)	50%
**PPV (95% CI)**	70.0% (45.7–88.1)	1) 100% (81–100) 2) 100% (88–100)	Not reported	Anemia: 76% (65–85) Polycythemia: 93% (93–100)	88%
**NPV (95% CI)**	99.3% (95.9–100)	1) 70% (58–80) 2) 88% (77–94)	Not reported	Anemia: 94% (83–98) Polycythemia: 99% (93–100)	33%

Abbreviations: AUC = area under the curve, Hb = hemoglobin concentration, MCA-PSV = middle cerebral artery peak systolic velocity, ΔMCA-PSV = difference in MCA-PSV between twins, MoM = multiples of the median, TAPS = twin anemia polycythemia sequence, wk = week.

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
