# Peer review of "Middle Cerebral Artery Doppler Velocimetry for the Diagnosis of Twin Anemia Polycythemia Sequence: A Systematic Review"

_jcm, 2020, doi:10.3390/jcm9061735_

Round 1
Reviewer 1 Report
This is a systematic review of the diagnostic efficacy of MCA-PSV in prenatal detection of TAPS including 5 studies. . Diagnostic criteria for the screening test and the diagnostic test were not consistently defined between the studies and therefore meta-analysis to estimate PPV, NPV, sensitivity and specificity was not performed.
The topic of prenatal detection of TAPS discussed in this paper is of great importance. The significance as well as the limitations in the diagnosis and management of TAPS are well presented.
A few issues should be addressed:
- The mean internal between the MCA-PSV measurement and fetal/ neonatal hemoglobin measurement should be reported for each study included
- The authors should discuss the value of MCA-PSV in prediction of fetal polycythemia as opposed to its value in prediction of fetal anemia as shown in the studies included in this review.
- Despite the limitations of the existing studies, the authors should emphasize the controversy regarding the best method for prenatal diagnosis of TAPS: absolute values vs relative difference between MCA-PSV of both twins- comparison of the performance of these two methods should be presented (see Ref # 10, Tollenaar et al, 2019)
- Is there any difference in the performance of prenatal detection of TAPS whether it is spontaneous or post laser? If available, data regarding this issue should be presented
-
Author Response
- The mean interval between the MCA-PSV measurement and fetal/neonatal hemoglobin measurement should be reported for each study included
Thank you for this important point. We agree that the interval between MCA-PSV measurement and hemoglobin measurement is important in determining the diagnostic efficacy of the MCA-PSV. Unfortunately, most of the studies in our review do not include this information directly. For De Sousa et al, Tollenaar et al, and Fishel-Bartal et al, twins were excluded from analysis if the difference between MCA-PSV and Hb measurement was greater than 1 week which establishes an upper bound. Veujoz et al. required the interval to be less than 48 hours. Only De Sousa reported the median value of the difference at 2 (0 – 7) days. We added a line in Table 4 reporting this data.
- The authors should discuss the value of MCA-PSV in prediction of fetal polycythemia as opposed to its value in prediction of fetal anemia as shown in the studies included in this review.
Thank you. We appreciated the importance of this point. We believe that this issue, and the next, are closely related and we have addressed them each by adding a new paragraph to the discussion. Please see below under #3.
- Despite the limitations of the existing studies, the authors should emphasize the controversy regarding the best method for prenatal diagnosis of TAPS: absolute values vs relative difference between MCA-PSV of both twins- comparison of the performance of these two methods should be presented (see Ref # 10, Tollenaar et al, 2019)
We have added the following paragraph to the 2nd paragraph of the discussion to make this point, as well as the polycythemia issue listed above:
“Of note, the work by Fishel-Bartal et al. highlights an emerging trend of favoring ΔMCA-PSV over absolute MCA-PSV values because of the former’s reported higher sensitivity19. Tollenaar et al. retrospectively observed that twins meeting ΔMCA-PSV criteria, but not absolute criteria, have similar postnatal outcomes to twins meeting absolute criteria. This led that group to propose a new staging system for TAPS based on ΔMCA-PSV10. Fishel-Bartal et al. suggest that the superiority of ΔMCA-PSV values may be related to the poor predictive ability of MCA-PSV < 1.0 MoM in diagnosing polycythemia, however Slaghekke et al. report high sensitivity and specificity using this method (Table 4)19,47. Further prospective data is required to validate (or invalidate) MCA-PSV < 1.0 MoM and ΔMCA-PSV in diagnosing polycythemia and TAPS respectively.”
- Is there any difference in the performance of prenatal detection of TAPS whether it is spontaneous of post laser? If available, data regarding this issue should be presented.
While there are case reports describing diagnoses of spontaneous and post-laser TAPS using MCA Doppler studies, we are not aware of any data that separates the two types of TAPS as part of a systematic evaluation of the diagnostic performance of MCA-PSV values. We have added a sentence in the Discussion (6th paragraph) to make sure this is clear; it now reads:
“Further work is required to provide a valid basis for the diagnosis and treatment of TAPS. sTAPS and plTAPs should be considered separately as data is sparse to suggest whether MCA-PSV values behave similarly in these two physiologically distinct scenarios. We are unaware of any data that separate the two entities as part of an evaluation of MCA-PSV as a screening test.”
Reviewer 2 Report
This manuscript describes a systematic review on middle cerebral artery doppler velocimetry for the diagnosis of TAPS.
The manuscript is well written, the method section is described in a clear and comprehensive way, and shows that the authors put a lot of effort in thoroughly analyzing and describing the articles. This systematic review demonstrates what is currently already believed (but now scientifically supported): there is no uniformity and consensus on MCA-PSV criteria for the diagnosis of TAPS.
The only comment that I have is that the authors should use consistent terminology to describe TAPS. In the title of the manuscript is says "twin anemia polycythemia SYNDROME" and in the main text the condition is referred to as "twin anemia polycythemia SEQUENCE". I would recommend using "sequence" as this is the most generally used term for TAPS.
Author Response
Thank you for the thoughtful review and for noticing the typo. We have made the correction in the title.